# Influence of Cavity Size on the Survival of Single Surface Atraumatic Restorative Treatment Using Glass Ionomer Cement with or without Chlorhexidine Diacetate—A Randomized Trial

Roshan Noor Mohamed [1], Sakeenabi Basha [2,*], Jooie S. Joshi [3] and Poornima Parameshwarappa [3]

1 Department of Pediatric Dentistry, Faculty of Dentistry, Taif University, P.O. Box 11099, Taif 21944, Saudi Arabia; roshan.noor@tudent.edu.sa
2 Department of Community Dentistry, Faculty of Dentistry, Taif University, P.O. Box 11099, Taif 21944, Saudi Arabia
3 Department of Pediatric Dentistry, College of Dental Sciences, Davangere 577004, India; drjooiejoshi@gmail.com (J.S.J.); drpoornimas@gmail.com (P.P.)
* Correspondence: sakeena@tudent.edu.sa; Tel.:+96-65-3841-8547

**Abstract:** The purpose of the present study was to assess the influence of cavity size on the survival of conventional and CHX modified GIC in single surface primary molars receiving Atraumatic Restorative Treatment (ART). A randomized controlled trial with a split-mouth design was conducted on 90 children with symmetrical bilateral single surface carious lesions on primary molars. The teeth were randomly allotted to the conventional GIC group (group 1, $n = 90$) and CHX modified GIC group (group 2, $n = 90$). Both groups received atraumatic restorative treatment under rubber dam isolation. The cavity size was measured in terms of depth, mesiodistal, and buccolingual dimensions. The survival of ART restorations was measured after 6, 12, 18, and 24 months. The difference in proportion was tested using the Kruskal–Wallis H test, and survival curve estimation was carried out using the Kaplan–Meier method. The overall survival of all ART restorations was 83.3% at 24 months for the total sample. The survival of conventional GIC at 24 months was 83.9%, and for CHX-modified GIC was 82.7% ($p > 0.05$). The collective overall success of 65.1% was seen in the cavity volume category of 10–29.9 mm$^3$. CHX modified GIC showed high survival percentage (60%) with depth >3 mm. To conclude, no significant difference was observed in the overall survival percentage of conventional and CHX modified GIC. Survival percentage was highest for cavities with a volume of 10–19.9 mm$^3$.

**Keywords:** ART; chlorhexidine; GIC; survival; primary teeth; single surface

## 1. Introduction

Atraumatic Restorative Treatment (ART) is a minimally invasive procedure that involves the sealing of pit and fissures and restoration of a cavitated carious lesion using Glass-Ionomer Cement (GIC) [1]. It was originally introduced for use in developing countries; however, it is now widely accepted globally as a treatment of choice because of its minimally invasive nature [2–4]. Recently, newer types of glass-ionomer cement have become available, such as high viscosity GIC's and Resin-Modified Glass-Ionomer Cement (RM-GIC's) with improved physical properties suitable for both single and multi-surface ART restorations [5–7]. However, Fuji IX glass-ionomer cement is still the most commonly used material for single-surface ART restorations [1,8]. The glass-ionomer used in ART has the advantage of chemical adhesion to the tooth structure, a good level of thermal biocompatibility with enamel and dentin, and remineralization of demineralized dentin [9]. To enhance the antimicrobial ability of GIC, few authors have incorporated chlorhexidine (CHX) into it [10–15]. However, the survival of conventional and or CHX modified GIC depends upon cavity size, number of surfaces involved, the experience of the operator, and isolation techniques used during the restorative procedure [16–24]. The cavity size



influences the survival of restoration in many ways [15,17,25]. The larger restorations have greater failure rates due to recurrent caries, fracture of restorations, and the larger surface area leading to restoration failure [15,17,24]. Previously, several studies [18,21,24,26–28] have focused on the survival of single surface ART restorations without considering the influence of cavity size. Although it is reported that physical properties are not affected in CHX modified GIC, [11,14,15] the survival of the same has been tested by a limited number of studies [10], especially in the primary dentition [15,24]. With this background, the present randomized trial was conducted to assess the influence of cavity size on the survival of conventional and CHX modified GIC in single surface primary molar teeth ART.

## 2. Materials and Methods

Study design, sample size, and ethical aspect: A triple blinded randomized controlled trial with the split-mouth design was conducted among 90 school children aged 5 to 9 years old. The study period was 24 months. The children included in the study were selected by using the simple random sampling lottery method from the outpatient clinic of the Department of Pedodontics and Preventive Dentistry, and after screening surrounding schools. Selected children belonged to a low to moderate socio-economic background with limited dental health care access. The sample size was determined using the parameters of previous ART systematic review and meta-analysis, which reported 94% survival for single surface primary posterior teeth ART restorations [23]. A clinically expected 12% survival difference between conventional GIC and CHX-modified GIC was considered with $\acute{a}$ error of 0.05, and power of 80%. A total of 75 single surface primary molars for each treatment group was required, which was rounded to 90 per group considering the dropouts (20%). Intraoral periapical radiographs were used to assess the pulpal involvement whenever necessary. Ethical clearance was obtained from the institutional review board (Protocol number- CODS/IRB/14/2017/01). Before treatment, written informed consent was obtained from all parents/ guardians.

The inclusion criteria to participate in the study are explained in Table 1.

**Table 1.** Inclusion and exclusion criteria for participants' enrolment in the study.

| | |
|---|---|
| Inclusion criteria | • Healthy participants without any history of medical condition or disease.<br>• Presence of at least two cavitated dentine carious lesions on the occlusal surface of primary molars situated on different sides of the jaw with a natural antagonist.<br>• Carious lesions were moderately deep, involving enamel and dentin, and accessible to hand excavation using ART procedure. |
| Exclusion criteria | • Definite or likely pulpal exposure or an associated abscess, pain or swelling, and adjacent soft tissue lesion.<br>• Teeth with enamel crack or fracture<br>• Teeth with a developmental disorder.<br>• Presence of abnormal oral habits. |

The material used: A split-mouth design was used for ART restorations with conventional GIC (GC Fuji IX, Tokyo, Japan) and modified GIC with chlorhexidine incorporation, assigned randomly to contralateral sides using the lottery method. Modified GIC was prepared by the addition of one wt/wt percent of CHX diacetate (RM1659-25G, HiMedia Laboratories, Mumbai, India) into the powder of the conventional glass ionomer (GC Fuji IX, Tokyo, Japan) by the geometric dilution method. One thousand mg of Fuji IX GIC was proportioned on a mixing pad and weighed, to which 10 mg (one percent) of CHX was added. Both the material was available in an identical container. CHX modified GIC was prepared by a non-operating dentist and labeled the container as I and II, respectively for conventional GIC and CHX modified GIC for effective blinding of the operating dentist.

Cavity size estimation: The size of the cavity was measured in terms of depth, mesiodistal, and buccolingual dimensions before the restoration of the cavity. The depth, mesiodistal width, and buccolingual width of the cavity were categorized into four groups (<2 mm, 2.1–3 mm, 3.1–4 mm, and >4 mm). The depth of the cavity at the deepest part was measured using a sterilized K-file with a rubber stopper and accurately recorded using a high-sensitivity dental caliper. The widest mesiodistal and buccolingual width of the cavity was directly measured using high sensitivity dental caliper after completion of cavity preparation. Cavity size volume was calculated and was divided into six groups (0–9.9 m$^3$, 10–19.9 m$^3$, 20–29.9 m$^3$, 30–39.9 m$^3$, 40–49.9 m$^3$, and >50 m$^3$).

Follow-up and survival estimation: The children were treated by a single operator performing all the ART restorations in the pediatric dentistry clinic on a dental chair. Each child received two ART restorations: one restored with conventional GIC and another with CHX incorporated GIC. The survival of ART restorations was assessed after 6, 12, 18, and 24 months using the code and criteria [28] summarized in Table 2.

**Table 2.** Code and criteria used for the assessment of ART restorations.

| Code | Criteria |
| --- | --- |
| 0 | The restoration is present and in good condition |
| 1 | The restoration is present, with a slight marginal defect; no repair is needed |
| 2 | The restoration is present, with slight wear; no repair is needed |
| 3 | The restoration is present, with marginal defect >0.5 mm; repair is needed |
| 4 | The restoration is present, with wear >0.5 mm; repair is needed |
| 5 | The restoration is not present, it is partly or completely lost |
| 6 | The restoration is not present, it is replaced by another restoration |
| 7 | The tooth is missing, exfoliated, or extracted |
| 8 | Restoration not assessed; child not present |

Codes: 0–2 = Successful; 3–6= Failure; 7–8= Excluded.

All the restorations were performed under rubber dam isolation. A blinded experienced dental doctor, who was not involved in the placement of restorations, evaluated the survival of ART restorations using sterilized plane mouth mirrors, World Health Organization (WHO) CPI periodontal probes, sharp sickle-shaped explorers, and a portable light source. The size of any marginal defect and the amount of wear was measured using the ball end of the CPI probe (0.5 mm in diameter).

A random 15% sample of children were subjected to duplicate examinations on each follow-up to assess the intra-examiner reproducibility; the overall Cohen's Kappa value in both assessments was 0.87.

Statistical analysis: The data were analyzed using a software program (SPSS 17.0 for Windows). The survival percentages of ART restorations depending on cavity size were analyzed using the Kruskal–Wallis H Test, followed by the Man-Whitney U test for intergroup comparison. The difference in the survival percentage of conventional and CHX modified GIC was analyzed using the Chi-Square test. Survival curve estimation was carried out using the Kaplan–Meier method. The log-rank test was used to check differences in survival percentage of conventional and CHX modified GIC restorations. All the tests were two-sided, and the difference was statistically significant if $p < 0.05$.

## 3. Results

A total of 90 children (average age of 6.8 ± 1.4 years, and male to female ratio 1.2:1), with 180 restorations (116 mandibular molars and 64 maxillary molars), were prepared. The consort flow diagram is explained in Figure 1.

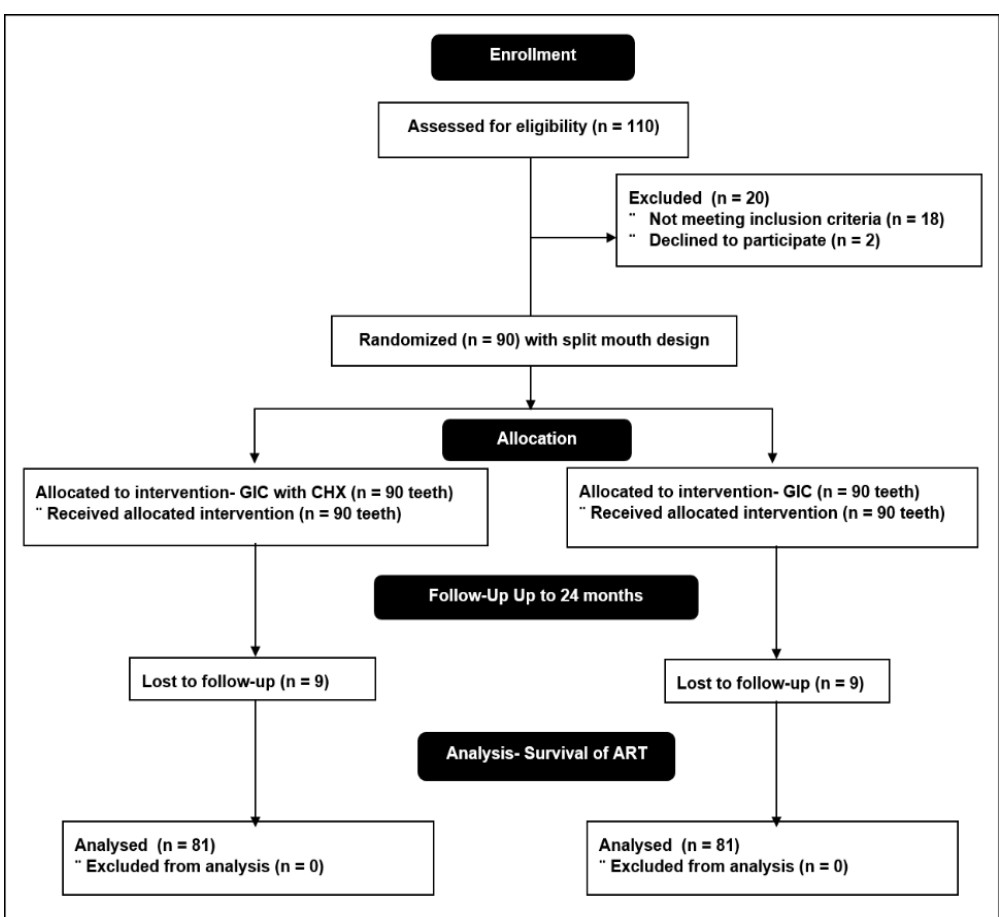

**Figure 1.** Consort statement flow diagram.

The distribution of cavity size in terms of depth, mesiodistal, and buccolingual dimensions is summarized in Table 3.

**Table 3.** Distribution of cavities according to the size at baseline.

| Cavity Sizes | Depth *n* (Mean) | Cavity Size Categories Mesio-Distal *n* (Mean) | Bucco-Lingual *n* (Mean) |
|---|---|---|---|
| <2 mm | 68 (1.6 mm) | 38 (1.5 mm) | 51 (1.4 mm) |
| 2.1–3 mm | 97 (2.3 mm) | 65 (2.6 mm) | 68 (2.4 mm) |
| 3.1–4 mm | 15 (3.2 mm) | 44 (3.3 mm) | 37 (3.3 mm) |
| >4 mm | 0 | 33 (4.2 mm) | 24 (4.2 mm) |
| Total | 180 | 180 | 180 |

The dropout rate for 12 months and the 24-month assessment was 4.4% and 10%, respectively. The overall survival of all ART restorations was 83.3% at 24 months for the total sample. The survival of conventional GIC, at 24 months assessment was 83.9%, and for CHX-modified GIC it was 82.7% ($p > 0.05$) (Table 4).

**Table 4.** Survival status of conventional GIC and CHX modified GIC ART restoration after 24 months.

| | Restoration Status | 24 Months GIC 81 | % | CHX-GIC 81 | % | Kruskal–Wallis *p* |
|---|---|---|---|---|---|---|
| 1. | Success, in good condition | 51 | 63.0 | 46 | 56.8 | 0.07 |
| 2. | Success, slight marginal defect | 8 | 9.9 | 13 | 16.0 | 0.09 |
| 3. | Success, slight wear | 9 | 11.1 | 8 | 9.9 | 0.09 |
| 4. | Failed, gross marginal defect | 5 | 6.2 | 6 | 7.4 | 0.11 |
| 5. | Failed, gross wear | 4 | 4.9 | 4 | 4.9 | 0.12 |
| 6. | Failed, a restoration partly or completely missing | 3 | 3.7 | 3 | 3.7 | 0.11 |
| 7. | Failed, restoration replaced by another filling | 1 | 1.2 | 1 | 1.2 | NA |
| Success | | 68 | 83.9 | 67 | 82.7 | 0.12 |
| Failure | | 13 | 16.0 | 14 | 17.3 | 0.11 |
| Overall success | | | | 135 (83.3) | | |
| Drop-out | | | | 9 | | |

GIC—Glass ionomer cement, CHX—Chlorhexidine, ART—Atraumatic restorative treatment.

There was a statistically significant difference in survival of ART restorations between the 6-month assessment and 24-month assessment (*p* = 0.03) for both conventional GIC and CHX Modified GIC. The most successful restorations were assessed to be in good condition (code-0) for both the groups, while the reason for failure was recorded maximum under gross marginal defect (code-3) (Table 4).

Survival of ART restorations depending on cavity size showed the highest success for restorations with 2.1–3 mm cavity depth, mesiodistal, and buccolingual width (Table 5, Figure 2).

**Table 5.** Survival Status of GIC, and CHX–GIC ART restorations based on cavity size at 24 months.

| Cavity Size | GIC Success | CHX GIC Success | Chi-Square, *p*-Value |
|---|---|---|---|
| Cavity Depth success | | | |
| a. <2 mm (*n* = 46) | 24 (52.2) | 22 (47.8) | 0.08 |
| b. 2.1–3 mm (*n* = 79) | 40 (50.6) | 39 (49.4) | 0.09 |
| c. 3.1–4 mm (*n* = 10) | 4 (40) | 6 (60) | 0.03 |
| Mesio-distal width, success | | | |
| <2 mm (*n* = 21) | 10 (47.6) | 11 (52.4) | 0.07 |
| 2.1–3 mm (*n* = 55) | 27 (49.1) | 28 (50.9) | 0.09 |
| 3.1–4 mm (*n* = 35) | 17 (48.6) | 18 (51.4) | 0.08 |
| >4 mm (*n* = 24) | 14 (58.3) | 10 (41.7) | 0.06 |
| Bucco-lingual width, success | | | |
| <2 mm (*n* = 39) | 18 (46.2) | 21 (53.8) | 0.06 |
| 2.1–3 mm (*n* = 62) | 33 (53.2) | 29 (46.8) | 0.07 |
| 3.1–4 mm (*n* = 20) | 9 (45) | 11 (55) | 0.06 |
| >4 mm (*n* = 14) | 8 (57.1) | 6 (42.9) | 0.06 |

GIC—Glass ionomer cement, CHX—Chlorhexidine, ART—Atraumatic restorative treatment.

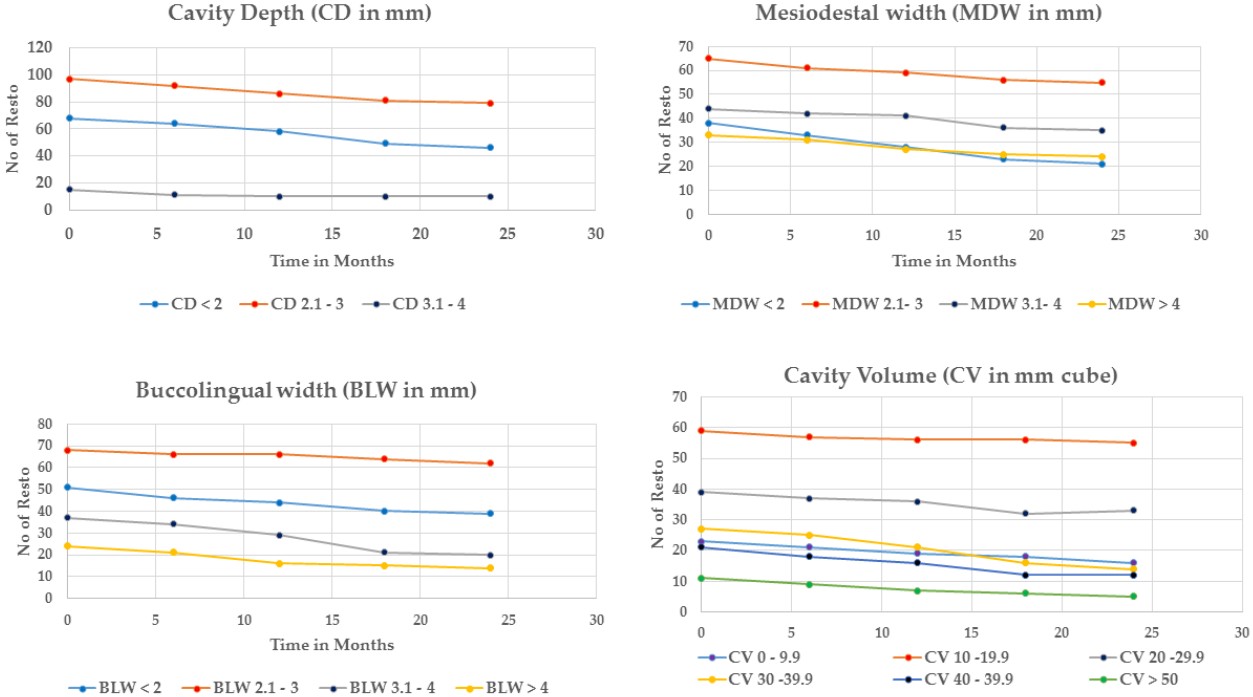

**Figure 2.** Overall survival of ART restorations according to cavity size.

The success percentage of ART restorations with cavity depths of 1.2 to 3 mm was 86.8%, which was significantly higher than cavity depth > 3 mm (71.4%). The survival success for the mesiodistal width of the cavity showed maximum success for the 2.1–3 mm (96.5%) category compared to the <2 mm category (60%). This finding was statistically significant ($p = 0.03$). For the buccolingual dimension of the cavity, the success of survival at 24-month assessment was 62.5% and 63.8%, respectively for cavity dimensions 3.1–4 mm and > 4mm when compared to the highest recorded success for 2.1–3 mm category, which was 98.4% ($p = 0.03$).

CHX modified GIC showed high survival percentage (60%) with depth >3 mm ($p = 0.03$) compared to conventional GIC. There was no statistical significance between GIC and CHX-GIC success for mesiodistal and buccolingual cavity width ($p > 0.05$).

The cavity size distribution based on the cavity volume at baseline and the survival of ART restorations at 24-month assessment is summarized in Table 6. The collective overall success of 65.1% was seen in the cavity volume category of 10–29.9 mm$^3$ (Table 6).

**Table 6.** Distribution of cavities on cavity volume at baseline and survival at 24 months.

| Cavity Volume Category | Cavity at Baseline ($n = 180$). n (Mean Volume) | Survival of ART Restorations at the 24-Month Assessment | | | | | |
|---|---|---|---|---|---|---|---|
| | | Overall Survival of ART Restorations ($n = 162$) | | Conventional GIC ($n = 81$) | | CHX–Modified GIC ($n = 81$) | |
| | | Success ($n = 135$) n (%) | Failure ($n = 27$) n (%) | Success ($n = 68$) n (%) | Failure ($n = 13$) n (%) | Success ($n = 67$) n (%) | Failure ($n = 14$) n (%) |
| a. 0–9.9 mm$^3$ | 23 (7.9) | 16 (11.8) | 5 (18.5) | 9 (13.2) | 3 (23.1) | 7 (10.4) | 2 (14.3) |
| b. 10–19.9 mm$^3$ | 59 (18.4) | 55 (40.7) | 2 (7.4) | 25 (36.8) | 0 | 30 (44.8) | 2 (14.3) |
| c. 20–29.9 mm$^3$ | 39 (26.2) | 33 (24.4) | 1 (3.7) | 19 (27.9) | 0 | 14 (20.9) | 1 (7.1) |
| d. 30–39.9 mm$^3$ | 27 (37.3) | 14 (10.4) | 8 (29.6) | 8 (11.8) | 4 (30.8) | 6 (8.9) | 4 (28.6) |
| e. 40–49.9 mm$^3$ | 21 (43.4) | 12 (8.9) | 7 (25.9) | 5 (7.3) | 3 (23.1) | 7 (10.4) | 4 (28.6) |
| f. >50 mm$^3$ | 11 (61.4) | 5 (3.7) | 4 (14.8) | 2 (2.9) | 3 (23.1) | 3 (4.5) | 1 (7.1) |
| Kruskal–Wallis H test, *p* value | | 0.03 | 0.08 | 0.03 | 0.07 | 0.03 | 0.09 |
| Man-Whitney U test | | b > a, c, d, e, f | NA | b > a, c, d, e, f | NA | b > a, c, d, e, f | NA |

GIC–Glass ionomer cement, CHX–Chlorhexidine, ART–Atraumatic restorative treatment.

Survival percentage with standard error for conventional and CHX modified GIC ART restoration at different time intervals is presented in Table 7.

**Table 7.** Survival percentage with standard error for conventional and CHX modified GIC ART restoration at a different time interval.

| Time Interval (Months) | GIC | | | | | CHX–GIC | | | | |
|---|---|---|---|---|---|---|---|---|---|---|
| | n [e] | n [f] | n [c] | Survival % | SE | n [e] | n [f] | n [c] | Survival % | SE |
| 0–6 | 90 | 3 | 3 | 96.6 | 1.7 | 90 | 4 | 3 | 95.4 | 1.9 |
| 6–12 | 86 | 8 | 4 | 90.7 | 2.1 | 86 | 9 | 4 | 89.5 | 2.3 |
| 12–18 | 83 | 12 | 7 | 85.5 | 3.1 | 83 | 14 | 7 | 83.1 | 3.6 |
| 18–24 | 81 | 13 | 9 | 83.9 | 3.8 | 81 | 14 | 9 | 82.7 | 3.9 |

n [e]—Teeth at entry, n [f]—Cumulative failure teeth, n [c]—Cumulative censored data, SE—Standard error, GIC—Glass ionomer cement, CHX—Chlorhexidine, ART—Atraumatic restorative treatment.

No significant difference was observed between the survival of conventional and CHX modified GIC ART restoration at different time intervals.

## 4. Discussion

Atraumatic restorative treatment is one of the minimally invasive procedures for restoration of carious lesions, which is well received because of its atraumatic nature, and ease of instrumentation without provoking much anxiety, especially in children [29]. The present study was conducted to assess the influence of cavity size on the survival of conventional and CHX modified GIC in single surface primary molar teeth ART. The result showed a cumulative survival rate of all ART restorations after a two-year follow-up was 83.3%. The recent systematic review by de Amorim et al. [23] showed survival of 94.3% for single surface posterior teeth ART with a 2-year follow-up. However, the systematic review used studies with conventional GIC. In the present study, both conventional, and CHX modified GIC was used. Duque et al. [15] showed an overall survival of 48% in multiple surface primary teeth restoration for both conventional and CHX modified GIC after a 1-year follow-up.

The current result showed no significant difference in the overall success of conventional (83.9%), and CHX modified GIC (82.7%) after 24 months follow-up. The result is in agreement with previous single and multiple surface ART restorations on primary and permanent teeth with conventional and CHX modified GIC [15,18]. This suggests that the addition of CHX did not alter the physical properties of GIC [11,14,15]. In the present study, the survival percentage decreased with age from 6 months to 24 months in both groups. This may be attributed to the cumulative effects of dropouts and failure of restoration through the 2-year follow-up period.

The cavity size plays an important role in the survival of restorations [15,17]. In the present study, cavities with a volume of 10–19.9 mm$^3$ showed the highest survival both in conventional and CHX modified GIC. The smallest and largest restorations showed a relatively poor survival percentage. This may be due to deficiency in the material in the smallest restorations and bulk failure or pulpal reactions in the largest restorations [15].

With cavity depth, the highest success was observed with a cavity depth of 2.1–3 mm both in conventional and CHX modified GIC ($p < 0.05$). At depth >3mm, the survival percentage was high for CHX modified GIC. This may be due to the enhanced antimicrobial ability of CHX modified GIC, which reduced pulpal infection in deep cavities and chances of failure [10,13]. Concerning buccolingual and mesiodistal width, the highest success was observed with a 2.1–3 mm width. High failure in the smallest and greatest width may be due to deficient material or bulk failure, respectively.

In the present study, the survival of ART was assessed clinically, using ART restoration criteria with codes from 0 to 8 [28]. This criterion has the advantage of assessing the marginal integrity of the restoration [30]. The most successful restorations were assessed to

be in good condition (code-0) for both the groups, while the reason for failure was recorded maximum under gross marginal defect (code-3). This is in line with previous studies, which showed the main reason for failure was due to a marginal defect in restoration [15,17,18].

The strength of the present study is the standardized procedure used, where all the restorations were carried out by a single operator, under the rubber dam isolation method. The split-mouth design was followed, with both conventional and CHX modified GIC restorations evaluated under the same caries risk and same individual conditions. The operator, evaluator, and statistician were all blinded concerning the application, evaluation, and analysis criteria, which reduced the risk of bias in the present study. As a future scope of the present study, newer GIC materials can be tested for success, based on cavity dimensions on a larger sample size of both single and multi-surface ART restorations.

## 5. Conclusions

The present study showed no significant difference in the overall survival percentage of conventional and CHX modified GIC. Survival percentage was highest for cavities with a volume of 10–19.9 mm$^3$. CHX modified GIC showed high survival percentage with depth > 3 mm compared to conventional GIC restorations. The main reason for failure was due to gross marginal defect.

**Author Contributions:** Conceptualization, R.N.M., S.B., J.S.J., and P.P.; methodology: R.N.M., S.B., J.S.J., and P.P.; software; R.N.M., J.S.J., S.B.; validation, R.N.M., S.B., J.S.J., and P.P.; formal analysis, R.N.M.; investigation, R.N.M., S.B., J.S.J., and P.P.; resources, R.N.M., S.B., J.S.J., and P.P.; data curation, R.N.M., S.B., J.S.J., and P.P.; writing—original draft preparation, R.N.M., S.B., J.S.J., and P.P.; writing—review and editing, R.N.M., S.B., J.S.J., and P.P.; visualization, R.N.M.; supervision, R.N.M., P.P.; project administration, R.N.M., J.S.J.; funding acquisition, R.N.M. All authors have read and agreed to the published version of the manuscript.

**Funding:** This research received no external funding.

**Institutional Review Board Statement:** The study was conducted according to the guidelines of the Declaration of Helsinki and approved by the Institutional Review Board (or Ethics Committee) of College of Dental Sciences (protocol code–CODS/IRB/14/2017/01).

**Informed Consent Statement:** Informed consent was obtained from all subjects involved in the study.

**Data Availability Statement:** Data will be made available as per request from corresponding author.

**Acknowledgments:** Present research work is supported by Taif University Researchers Supporting Project number (TURSP-2020/102), Taif University, P.O. Box 11099, Taif 21944, Saudi Arabia.

**Conflicts of Interest:** The authors declare no conflict of interest.

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
