# Peer review of "Influence of Cavity Size on the Survival of Single Surface Atraumatic Restorative Treatment Using Glass Ionomer Cement with or without Chlorhexidine Diacetate—A Randomized Trial"

_applsci, doi:10.3390/app112110438_

Round 1

Reviewer 1 Report

Excellent meticulous analysis of cavity size on restoration of carious teeth.

Recommend publication after minor corrections, e.g. Line 16: Add period; Line 17; Add a verb to complete sentence.

Author Response

Response to Reviewer 1 Comments

Point 1: English language and style are fine/minor spell check required
Response 1: English language and grammar corrections were done using Grammarly soft ware

Point 2: Recommend publication after minor corrections, e.g. Line 16: Add period; Line 17; Add a verb to

complete sentence.

Response 2: Necessary corrections were done in line 16 and 17 as per the suggestion of reviewer.

Reviewer 2 Report

First, I would like to congratulate the authors for the article with an interesting topic, but I am afraid the structure of the article should be improved in order to publish it. Other important thing is the references; you should review number 23 and 24 both are in different editing.

In the abstract, you should have the same structure as your article, and the sentences should be complete, you are missing some words, I assume that you do so to have the number of words correctly, but in the abstract, you do not need to explain ALL your study. Therefore, you should re-write it with the same structure you have in the article.

In the introduction you use develop three times in two lines you should change the verb. ART was first introduced in the 70´s in developing countries, and now is a technique used worldwide, but with more dental materials than the GIC, now the dental industry has developed other materials that has better properties (AAPD Policy on Alternative Restorative Treatment (ART) Pediatr. Dent. 2014;29:38.) You should explain the different types of materials used nowadays. I suggest you give more background with more articles regarding studies about cavity size using ART (Lo E., Holmgren C. Provision of atraumatic restorative treatment (ART) restorations to Chinese preschool children: A 30-month evaluation. Int. J. Pediatr. Dent. 2001;11:3–10. doi: 10.1046/j.1365-263x.2001.00232.x.)

In the material and methods paragraph, you should make some big changes. You mix the study design with part of the results, the average age, the baseline indexes should be in the results. I need an explanation on how you determined the sample size, based on a meta-analysis?? Should it be determined on your clinic patients? You should present you inclusion criteria, it is not clear.

Your study is based in comparing two types of GIC in different types of cavities, in terms of depth; I should change the structure of the material paragraph because I think it is all mixed.

Going with the results you could edit properly the table 3, it is mixed with the number of the lines.

I suggest you to re-write paragraph from line 159 to line 167, you mix the success rate of the ART according with its length, but you also say that the CHX modified GIC had more success rate than GIC group… I should write in one paragraph the success rate comparing the depth of the cavities, and in other paragraph I would compare the success rate with GIC and CHX modified GIC.

Author Response

Response to Reviewer 2 Comments

Point 1: Moderate English changes required

Response 1: English language and grammar corrections were done using Grammarly soft ware

Point 2: First, I would like to congratulate the authors for the article with an interesting topic, but I am afraid the structure of the article should be improved in order to publish it. Other important thing is the references; you should review number 23 and 24 both are in different editing.

Response 2: structure of the article improved as per the suggestions and reference number 23 and 24 modified as per the suggestion.

Point 3: In the abstract, you should have the same structure as your article, and the sentences should be complete, you are missing some words, I assume that you do so to have the number of words correctly, but in the abstract, you do not need to explain ALL your study. Therefore, you should re-write it with the same structure you have in the article

Response 3: Abstract modified and re-written as per the suggestion.

Point 4: In the introduction you use develop three times in two lines you should change the verb. ART was first introduced in the 70´s in developing countries, and now is a technique used worldwide, but with more dental materials than the GIC, now the dental industry has developed other materials that has better properties (AAPD Policy on Alternative Restorative Treatment (ART) Pediatr. Dent. 2014;29:38.) You should explain the different types of materials used nowadays. I suggest you give more background with more articles regarding studies about cavity size using ART (Lo E., Holmgren C. Provision of atraumatic restorative treatment (ART) restorations to Chinese preschool children: A 30-month evaluation. Int. J. Pediatr. Dent. 2001;11:3–10. doi: 10.1046/j.1365-263x.2001.00232.x.)

Response 4: Introduction modified with inclusion of new restorative materials, and how the cavity size influences the ART survival as per the suggestion

Point 5: In the material and methods paragraph, you should make some big changes. You mix the study design with part of the results, the average age, the baseline indexes should be in the results. I need an explanation on how you determined the sample size, based on a meta-analysis?? Should it be determined on your clinic patients? You should present you inclusion criteria, it is not clear. Your study is based in comparing two types of GIC in different types of cavities, in terms of depth; I should change the structure of the material paragraph because I think it is all mixed.

Response 5: The result is removed from material and method section. Sample size estimation explained as per the suggestion. Inclusion and exclusion criteria explained in Table 1. The structure of the material and method section modified with inclusion of material and cavity size in different paragraphs.

Point 6: Going with the results you could edit properly the table 3, it is mixed with the number of the lines.

Response 6: The Table edited as per the suggestion.

Point 7: I suggest you to re-write paragraph from line 159 to line 167, you mix the success rate of the ART according with its length, but you also say that the CHX modified GIC had more success rate than GIC group… I should write in one paragraph the success rate comparing the depth of the cavities, and in other paragraph I would compare the success rate with GIC and CHX modified GIC.

Response 7: The result modified with inclusion of separate paragraph for comparing depth of cavities and success rate with GIC and CHX modified GIC.

Reviewer 3 Report

Accept after minor revision

Congratulations on your work which, I found interesting.

Manuscript: Influence of cavity size on the survival of single surface Atraumatic restorative treatment using glass ionomer cement with or without chlorhexidine diacetate- a randomized trial., it is well written with an adequate structure as a scientific paper demands.

I have some minor revisions to propose to you to improve your work. Please refer to the following comments:

Line 117 I think it’s a typo, but you write “180 restoration (166 mandibular molars and 64 maxillary molars)” the sum of 166 and 64 is 230 not 180

Figure 2 is illegible. Please correct it so that you can read the presented data

In the „discussion” - Please indicate a further research plan and limitation of the study

The literature is old, with many citations of works from before 2012 - please consider updating it.

Author Response

Response to Reviewer 3 Comments

Point 1: Line 117 I think it’s a typo, but you write “180 restoration (166 mandibular molars and 64 maxillary molars)” the sum of 166 and 64 is 230 not 180.

Response 1: The typo error corrected (116 mandibular molars and 64 maxillary molars)” the sum of 116 and 64 is 180.

Point 2: Figure 2 is illegible. Please correct it so that you can read the presented data.

Response 2: The figure 2 modified as per the suggestion.

Point 3: In the „discussion” - Please indicate a further research plan and limitation of the study

Response 3: indicated a further research plan and limitation of the study as per the suggestion.

Point 4: The literature is old, with many citations of works from before 2012 - please consider updating it.

Response 4: The literature updated as per the suggestion.

Round 2

Reviewer 2 Report

Congratulations, the improvements that you made on the article are great. Now it has the structure you can expect on a scientific article.

Yet there still are some points of improvement that you should review in order to have it perfect:

The abstract present two different font types, with different size, please check that.

The introduction is ok.

In material and methods, now it is very clear how you chose your sample size. The inclusion and exclusion criteria should be reviewed, now it is clear but you should not entry a negation on inclusion criteria, I mean, if you say without abnormal oral habits you sholud stay in exclusion criteria abnormal oral habits. 

Author Response

Response to Reviewer 2 Comments

Point 1: Moderate English changes required

Response 1: English language and grammar corrections were done using Grammarly soft ware

Point 2: The abstract present two different font types, with different size, please check that.

Response 2: Uniform font type and size was corrected as per the suggestion

Point 3: In material and methods, now it is very clear how you chose your sample size. The inclusion and exclusion criteria should be reviewed, now it is clear but you should not entry a negation on inclusion criteria, I mean, if you say without abnormal oral habits you should stay in exclusion criteria abnormal oral habits.

Response 3:  Abnormal oral habits moved to exclusion criteria as per the suggestion